# Effects of Dietary Supplementation with Protected Sodium Butyrate on Gut Microbiota in Growing-Finishing Pigs

**DOI:** 10.3390/ani11072137

**Published:** 2021-07-19

**Authors:** María Bernad-Roche, Andrea Bellés, Laura Grasa, Alejandro Casanova-Higes, Raúl Carlos Mainar-Jaime

**Affiliations:** 1Departamento de Patología Animal, Facultad de Veterinaria, Instituto Agroalimentario de Aragón-IA2, Universidad de Zaragoza-CITA, 50013 Zaragoza, Spain; mbernadroche@gmail.com (M.B.-R.); acasanova@unizar.es (A.C.-H.); 2Departamento de Farmacología y Fisiología, Facultad de Veterinaria, Instituto Agroalimentario de Aragón-IA2, Universidad de Zaragoza-CITA, 50013 Zaragoza, Spain; a.belles@unizar.es (A.B.); lgralo@unizar.es (L.G.)

**Keywords:** pig, microbiota, intestinal microbiome, sodium butyrate, organic acid

## Abstract

**Simple Summary:**

The addition of protected sodium butyrate to the diet of fattening pigs during the whole fattening period (≈90 days) at a dose of 3 kg per ton of feed, did not modify the overall richness of microbiota composition of the pigs at slaughter, but may have caused some significant changes in specific taxa that could be associated with better gut health parameters. In any case, these results should be taken with caution, as the role of a given *taxon* on the pig’s gut health is influenced by numerous variables such as age, diet, environment, treatments, other taxa present, infections, or even the physiological status of the animal.

**Abstract:**

The study assessed changes in the gut microbiota of pigs after dietary supplementation with protected sodium butyrate (PSB) during the growing-fattening period (≈90 days). One gram of colon content from 18 pigs (9 from the treatment group -TG- and 9 from the control group -CG-) was collected. Bacterial DNA was extracted and 16S rRNA high-throughput amplicon sequencing used to assess microbiota changes between groups. The groups shared 75.4% of the 4697 operational taxonomic units identified. No differences in alpha diversity were found, but significant differences for some specific taxa were detected between groups. The low-represented phylum *Deinococcus-Thermus*, which is associated with the production of carotenoids with antioxidant, anti-apoptotic, and anti-inflammatory properties, was increased in the TG (*p* = 0.032). *Prevotellaceae*, *Lachnospiraceae*, *Peptostreptococcaceae*, *Peptococcaceae*, and *Terrisporobacter* were increased in the TG. Members of these families have the ability to ferment complex dietary polysaccharides and produce larger amounts of short chain fatty acids. Regarding species, only *Clostridium butyricum* was increased in the TG (*p* = 0.048). *Clostridium butyricum* is well-known as probiotic in humans, but it has also been associated with overall positive gut effects (increased villus height, improved body weight, reduction of diarrhea, etc.) in weanling pigs. Although the use of PSB did not modify the overall richness of microbiota composition of these slaughter pigs, it may have increased specific taxa associated with better gut health parameters.

## 1. Introduction

The study and implications of gut microbiota on pig health has been the objective of many recent studies, and an overall correlation between microbiota and health has been observed [1,2,3]. Gut microbiota is involved in many physiological functions, one of the most important being the digestion of nutrients. Commensal bacteria split long carbohydrate chains into short chain fatty acids (SCFA), such as acetic, propionic, and butyric acid, and these SCFA are used by the epithelium cells to produce energy [4]. In addition, intestinal bacteria may also play an essential role in disease prevention through the maintenance of the appropriate structure of the intestinal wall and by exerting some regulation of the immune function [3]. Thus, the loss of the gut microbial ecosystem diversity has shown to be related to an increase of the risk of gastrointestinal diarrhea and some immune-mediated diseases in post-weaning pigs [5], and even to functional implications on brain development and behavior [6]. Therefore, animal health and welfare could be improved by selection, nutrition, and management processes that take into account the role of the gut microbiota [7].

The intestinal microbiota is a non-static, complex structure that can be easily altered by many factors [8]. Animal factors such as age, health status, diet, breed, or the environment where the animal lives are responsible for changes in the gut microbiota. It also varies in each section of the intestine and, consequently, in the sampling location, being these differences either qualitative or quantitative [9]. Most of the commensal and beneficial bacteria belong to the *Firmicutes* and *Bacteroidetes* phyla, which are the predominant bacteria in healthy pigs (85% to 90% of the population). Other important but less frequent phyla are *Proteobacteria*, which includes the *Enterobacteriaceae* family, *Actinobacteria*, *Spirochaetes*, and *Verrucomicrobia*.

Among the major genera within these two main phyla would be *Clostridium*, *Lactobacillus*, *Lactococcus*, *Streptococcus*, *Blautia*, *Ruminococcus*, *Roseburia*, and *Enterococcus* (*Firmicutes*), and *Bacteroides* and *Prevotella* (*Bacteroidetes*), suggesting a core microbiota for healthy commercial swine [9,10]. Relative increases in the proportion of *Bacteroidetes* (genus *Prevotella*), *Firmicutes* (genera *Veillonela*, *Enterococcus*), *Proteobacteria* (*E. coli*, *Campylobacter* spp.) have been associated with diarrhea in the first weeks of piglets’ life [11,12,13,14,15]. In other recent study were the genera Escherichia and Shigella the core component of diarrheic microbiota, while Prevotella was characteristic of non-diarrheic microbiota of piglets [16]. These variable results show the difficulties in understanding the complex microbial relationships occurring within the intestinal tract.

Organic acids (OA) are being used as an alternative to antibiotics in pigs for the control of digestive disorders caused by enteric pathogens such as *Salmonella* or enterotoxigenic *E. coli*. Several effects have been associated with the administration of OA that would promote intestinal health. For instance, OA, such as butyric acid, are the preferred energy substrate for colon cells, promoting their normal proliferation and differentiation [17]. It also stimulates mucus production and intestinal secretions with the subsequent benefit in intestinal health, and acts as a powerful inhibitor of the intestinal inflammation and of the development of tumors [18]. OA seem also to modulate the immune response of the animals [19].

However, medium chain fatty acids, such as lauric acid or sodium heptanoate, and short chain fatty acids, such as formic or butyric acid, may also positively modify microbiota composition [20,21,22]. There are two major mechanisms by which they influence the intestinal microbiota, namely, (i) by reducing the pH of the intestinal environment; and (ii) by penetrating the pathogenic bacterial cell and altering its physiological homeostasis [23]. A reduction of the intestinal pH leads to the modification of the microbial composition as it selects for commensal acid-resistant microbes, such as lactic acid bacteria. Besides, enteric pathogens are usually sensitive to a low pH, and in some cases (butyric), it helps to down-regulate some invasion genes of *Salmonella*. In addition, OA would enter the pathogenic bacteria cell in their non-dissociated form and decrease the intracellular pH when dissociating, thus disrupting DNA synthesis [24,25]. Altogether reducing the overall proportion of pathogenic bacteria [26].

Most of the studies on the effects of OA on the modification of the intestinal microbiota in swine have been carried out in young pigs, i.e., suckling and/or weaned piglets, and after administration of OA for a limited period of time. However, few studies have focused on older pigs or after long treatments. Thus, the aim of this study was to assess changes in the gut microbiota of healthy fattening pigs after dietary supplementation with a formula of protected sodium butyrate (PSB) during the whole growing-fattening period.

## 2. Materials and Methods

### 2.1. Animals, Treatment and Sample Collection

Selected pigs (a mixed breed of Landrace -25%-, Large White -25%-, and Pietrain -50%-) were raised in a small commercial fattening unit (≈100 pigs in 8 pens; 12–14 pigs/pen). During the fattening period, the animals were fed with three different diets (starting, fattening and finishing) free of antibiotics (Appendix A). Drinkable water and feed were offered *ad libitum* for all the pigs. Animals were raised following European animal welfare regulations for pig farms (COUNCIL DIRECTIVE 2001/88/EC of 23 October 2001).

A feed additive (GUSTOR BP70, Norel S.A., Madrid, Spain) was administered to animals from 4 randomly selected pens (treatment group -TG-) along the whole period. This feed additive is a form of sodium butyrate (70%), part of which was protected with vegetable fat (hydrogenated palm stearin, 30%) in order to be able to reach the distal part of the intestinal tract, according to manufacturer. The remaining 4 pens were fed with the same diets without the additive (control group -CG-).

Piglets entered into the growing-fattening unit at 10 weeks of age and remained there until slaughter, that is, when they reached an average live weight of 100 ± 10 kg (≈6 months old). The treatment began 15 days after pigs entering into the fattening unit and a dose of 3 kg of the protected sodium butyrate (PSB) per ton of feed was used for the whole period, and for each of the three different diets that were administered. At slaughter, intestinal content from the colon of 18 randomly selected pigs (9 from the TG and 9 from the CG) was collected for laboratory analysis.

### 2.2. Bacterial DNA Extraction

One gram of colon content was collected and immediately frozen at −80 °C until processed. Bacterial DNA was extracted from frozen samples using a QIAamp Fast DNA Stool Mini Kit (Qiagen, Hilden, Germany) and following the manufacturer’s instructions. Fecal samples were mixed with 1 mL InhibitEX buffer in SK38 tubes and processed by using the Precellys^®^ 24 homogenizer for 2 × 30 s at 6500 rpm and 10 s delay between cycles. Lysis was completed at 95 °C for 5 min. Finally, DNA was eluted in 40 μL elution ATE buffer. Once DNA was extracted, DNA concentrations were measured with Qubit^®^ 4.0 fluorometer (Invitrogen) and dsDNA HS (high sensitivity) Assay Kit (Invitrogen). DNA purity was assessed by measuring the A260/A280 with NanoDrop^®^ ND-1000 Spectrophotometer V3.0.1 (Thermo Scientific, Waltham, MA, USA) and monitored on 1% agarose gels.

### 2.3. Library Preparation

Depending to the concentration, DNA was diluted to 1 ng/μL using sterile water. 16S rRNA gene of V3-V4 region was amplified using specific primers (515F-806R) [27] with a barcode. PCR reactions were carried out with Phusion^®^ High-Fidelity PCR Master Mix (New England Biolabs). 1X loading buffer (contained SYBR green) was mixed with PCR products and amplicons were detected by electrophoresis on 2% agarose gel. Samples with a bright main band between 400–450 bp were chosen for further experiments. PCR products were mixed in equidensity ratios. Then, the mixture of PCR products was purified with Qiagen Gel Extraction Kit (Qiagen, Hilden, Germany).

### 2.4. 16S rRNA Gene Sequencing

Sequencing libraries were generated using NEBNex Ultra DNA Library Pre^®^ Kit for Illumina, following manufacturer’s recommendations and index codes were added. The library quality was assessed on the Qubit^®^ 2.0 Fluorometer (Thermo Scientific) and Agilent Bioanalyzer 2100 system. Finally, the library was sequenced on an Illumina MiSeq platform at Novogene (Tianjin, China), and 250 bp paired-end reads were generated. Output data were stored in FASTQ format.

### 2.5. Bioinformatics and Statistical Analysis

Paired-end reads were assigned to samples based on their unique barcode and truncated by cutting off the barcode and primer sequence. These reads were merged using FLASH [28]. Quality filtering on the raw tags was performed under specific filtering conditions to obtain high-quality clean sequences [29] according to the QIIME 1.7.0 quality-controlled process [30]. The UPARSE software was used to pick up operational taxonomic units (OTUs) at 97% similarity [31]. For each OTU, the representative sequence was assigned to a taxonomic annotation by using the mothur program and SILVA SSU rRNA database [32,33].

Alpha diversity was used to measure within-groups microbial diversity (CG and TG). Simpson, Shannon, ACE, and Chao1 indexes were calculated with QIIME (Version 1.7.0) and displayed with R software (Version 2.15.3). The differences in the alpha diversity indexes between groups were analyzed by *t*-test, Wilcox, and Tukey tests (*p* < 0.05).

In order to estimate the dissimilarities between groups (beta diversity), both weighted and unweighted Unifrac were calculated by QIIME 1.7.0. Analysis of similarities (ANOSIM) and permutational multivariate analysis of variance (Adonis) were conducted using the vegan R package [34]. Differences in the weighted and unweighted Unifrac between the treatment and control groups were analyzed by *t*-test, Wilcox, and Tukey tests (*p* < 0.05).

Statistical analyses were performed to determine those taxa whose abundance varied significantly (*p* < 0.05) among groups. A *t*-test was carried out at various taxon ranks, including phylum, family, genus, and species. In addition, the microbial community differences between TG and CG diets were analyzed by the linear discriminant analysis (LDA) effect size (LEfSe) [35]. LEfSe method was used to identify the most differentially abundant taxa between the two groups, which would help to reveal potential microbial biomarkers.

## 3. Results

### 3.1. Sequencing Metrics

After quality control and chimera removal, the minimum and maximum of high-quality sequences in the samples were 101,778 and 191,444, being the average 157,755 sequences. The maximum percentage of chimeric sequences found in the samples was 17.8%, indicating a good performance of the sequencing.

The total number of OTUs identified was 4697, from which 3542 OTUs (75.4%) were shared by the CG and TG groups. Each sample had, on average, 2389 OTUs (range: 1928–2875). The rarefaction curves (not presented) showed asymptotic tendency indicating that sequencing depth was adequate for characterizing the pig gut microbiota in the present study.

### 3.2. General Description of Bacterial Communities

The gut microbiome dataset from the TG and CG was composed by more than 35 phyla, with four of them comprising almost 95% of all the bacteria (*Firmicutes* -62.0%-, *Bacteroidetes* -25.3%-, *Proteobacteria* -3.7%-, and *Spirochaetes* -3.2%-).

When comparing the abundance of the different taxa between the TG and the CG, a somewhat more abundant bacterium from the *Firmicutes* and *Verrucomicrobia* phyla were found in the TG compared to the CG, while those from the *Bacteroidetes* and *Spirochaetes* phyla were more abundant in the CG. However, no large differences were observed between groups at this level, neither at order and class levels. Major differences were found mostly when comparing families and genera (Figure 1).

The relative differences between the CG and TG for the 35 most common genera are presented in a heat map (Figure 2). There was a visible clustering of pigs according to treatment. The abundance of genera *Clostridium*, *Terrisporobacter*, *Alloprevotella*, and *Prevotella* increased mainly in TG, while it tended to decrease in CG. In contrast, the abundance of genera *Ruminococcus* and *Streptococcus* increased mainly in CG and decreased in TG.

### 3.3. Alpha Diversity

In Table 1 the Shannon, Simpson, ACE, and Chao1 indexes are shown for both groups. A high microbial diversity was observed for both groups as indicated by the Shannon and Simpson indexes, which were not significantly different between groups (Simpson index was close to 1 for both groups, and the Shannon index around 7). Neither significant differences were found in the Chao nor ACE indexes between both groups.

### 3.4. Beta Diversity

No statistically significant differences in the weighted and unweighted Unifrac analyses between the TG and the CG were found, but both the analysis of similarities (ANOSIM) and the permutational multivariate analysis of variance (Adonis) showed significant community differences between the CG and the TG (ANOSIM: R = 0.013, *p* = 0.042; Adonis: R^2^ = 0.11; *p* = 0.04). In Table 2 are shown the specific taxa that were significantly prevalent within each group when compared to the other according to *t*-test analysis.

According to the LEfSe analysis, the relative abundance of *Prevotellaceae* and *Peptostreptococcaceae* in the TG was significantly higher than in the CG, while the relative abundance of *Ruminococcaceae* and *Bacteroidales* S-24 group was significantly lower. The LEfSe cladogram as a result of the TG and CG is shown in Figure 3.

## 4. Discussion

Most studies focused on the impact that the use of PSB may have on the pig gut microbiota have been usually performed on young piglets (mostly from birth to post-weaning) treated for a short period of time [36,37,38,39]. There is only a single recent study on the effect of dietary sodium butyrate in a population of growing-finishing pigs, after being treated for a long period (69 days) [40]. However, in contrast to our study, the samples were collected from the caecum. Factors such as animals’ age, basal diet, as well as time of treatment or the site of sampling, have a major influence on microbiota composition [1,9,23,41,42,43]. Thus, this study can be considered unique as it was performed on growing-finishing pigs, the treatment was administered for a long period (>90 days) and samples were collected from the colon.

No significant differences in overall microbial diversity were detected between the CG and the TG. A high diversity was observed in both groups, as indicated by a Simpson index close to 1 and a Shannon index around 7, both being similar for the CG and TG (*p* > 0.05). These alpha diversity indexes suggested a similar evenness of this OTUs distribution. Chao and ACE indexes were also similar for both groups, meaning no differences in microbiota richness between the CG and the TG. Indeed, the CG and the TG shared more than 75% of the OTUs identified. It is interesting to note that the high alpha diversity values observed for both groups in our study were similar to those found in sows [1], suggesting an adult and therefore more stable microbiota composition [44], which would likely preclude finding major overall differences in biodiversity. Although in one previous study it was found a reduction of the microbiota diversity after sodium butyrate supplementation, it may have been related to the use of a reduced dose of two antibiotics, colistin and kitasamycin, along with the organic acid [40], as both antibiotics likely had some impact on specific microbial populations [41].

Therefore, the potential effects of PSB on fattening pigs should be then investigated at lower levels, i.e., specific taxa. The most abundant phyla found in both the CG and the TG were *Firmicutes* and *Bacteroidetes*, followed by *Proteobacteria*, in agreement with the core microbiota of the swine gut reported in previous studies [9,10]. Beta diversity measures showed a significant microbial community difference between both groups as indicated by results from the ANOSIM and Adonis, but the magnitude of these differences would not be large, as suggested by the corresponding R and R^2^ values (ANOSIM: R = 0.013, *p* = 0.042; Adonis: R^2^ = 0.11; *p* = 0.04). However, both the *t*-test and the LEfSe analyses detected specific taxa that differed from one group to the other. Thus, at the phylum level, we only found an increase of bacteria from the low-represented phylum *Deinococcus-Thermus* in the TG compared to the CG (*p* = 0.032; Table 2). This phylum is composed of the highly radioresistant order *Deinococcales* and the thermophilic order *Thermales*. Bacteria from this phylum are highly resistant to extreme stress through the production of carotenoids [45], which are known to have antioxidant, anti-apoptotic, and anti-inflammatory properties [46]. Thus, increasing numbers of these bacteria might favor a healthier gastrointestinal tract.

At a family level, members of the *Prevotellaceae*, *Lachnospiraceae*, *Peptostreptococcaceae*, *Peptococcaceae*, and *Terrisporobacter* families were increased in the TG (Table 2; Figure 2 and Figure 3). The first one being Gram-negative bacteria belonging to the order *Bacteroidales* while the others were Gram-positive *Clostridiales*. In contrast, *Ruminococcaceae* and the *Bacteroidales* S24-7 group were more abundant in the CG.

*Prevotella* has been described as the most abundant bacterium in the pig, although its relative abundance tends to decrease from weaning (25–50% of the bacteria) to finishing (10%) [47]. More than twenty species have been described within the genus *Prevotella* in the pig gastrointestinal tract, some of which were clearly more abundant in the pigs from the TG (Figure 2). Previous studies had already observed that members of the family *Prevotellaceae* were promoted when sodium butyrate was supplemented [21], suggesting that the dietary use of butyric acid would enhance their abundance.

Studies on the *Prevotella* genus have found contradictory results with regard to its relationship with pig feed efficiency and growth performance, maturation of mucosal immunity or interactions with other commensal bacteria. In general, *Prevotella* has been associated positively with feed efficiency and growth performance [48,49], likely due to their ability to ferment complex dietary polysaccharides, which would make larger amounts of energy available for the pig [50]. Moreover, *Prevotella* and *Ruminococcus* have been previously described as predominant and exclusive genera associated to two main enterotypes: the enterotype A in which it would predominate *Prevotella* and *Mitsuakella*; and enterotype B, with *Ruminococcus* and *Treponema*. Better growth traits have been found in those pigs with *Prevotella* predominant gut microbiota [48]. Interestingly, in our study, a significant reduction of the abundance of *Ruminococcaceae* was observed in the TG, but was increased in CG, supporting this co-exclusion effect. Since zootechnical parameters could not be measured in this study, the potential positive link between *Prevotella* predominant gut microbiota and favorable pig performance could not be studied.

*Prevotella* may also contribute to the maturation of mucosa immunity through the production of acetate, which is further used by other microbial species to produce butyrate. These SCFAs would participate in the communication between the microbiota and the gut immune system and help to maintain the anti/pro-inflammatory balance [51]. It appears that a gut environment with higher levels of butyric would favor the presence of this genus. In any case, most of these studies were performed on piglets and cannot be directly extrapolated to fattening pigs like the ones used in our study, as piglets are likely more prone to microbiota changes than older pigs with a more mature microbiota [42].

*Lachnospiraceae*, *Peptostreptococcaceae*, *Peptococcaceae*, and *Terrisporobacter* are a group of bacteria characterized by producing SCFA after degradation of indigestible plant-derived polysaccharides (i.e., cellulose and hemicellulose components). Within this group, *Lachnospiraceae* may be one of the most studied, with some members of this family showing strong hydrolyzing activities, being butyrate producers and acetate consumers, especially at a mildly acidic pH. Other *Lachnospiraceae* species and strains, however, produce formate, lactate, or H_2_ in addition to butyrate [52]. Studies on human microbiota have associated an increase of some of these genera (i.e., *Blautia*, which was also found increased in the TG in this study) and the production of butyrate with the control of gut inflammatory processes, atherosclerosis, and maturation of the immune system [53]. Members of this family have been also considered opportunistic pathogens associated with human disease, mostly with metabolic disease (i.e., obesity), irritable bowel syndrome [52], or even with high-fat diets [54,55,56], but these relationships may be irrelevant in a pig production context.

Less studied have been *Peptostreptococcaceae* and *Terrisporobacter*. The former appeared to be more prevalent in healthy than in disease animals. They have been inversely related to dysfunction of the intestinal barrier in rats [57], and positively associated with improved barrier ileal function in moderate protein diets in finishing pigs [58], thus suggesting they help to maintain gut homeostasis. Regarding *Terrisporobacter*, its abundance has been correlated with body weight, triglyceride, and worse serum lipid profile in elder women [59], thus being considered a potential obesity-promoting bacteria that may also affect sows [60], but it is unknown the relevance it may have on slaughter pigs.

Among the species taxon, only *Clostridium butyricum* was significantly increased in the TG (Table 2). It seems that the acidic environment generated by the PSB delivered to the distal part of the intestinal would have favored the presence of this bacterium. The positive effects of *C. butyricum* are well-known, and it has been used as a probiotic in cases of *Clostridioides difficile* infection or inflammatory bowel disease in humans [61,62]. It generates large amounts of butyrate through the fermentation of undigested dietary fibers and, as discussed before, this SCFA is associated with several beneficial effects, such as the improvement of the gastrointestinal barrier function, the modulation of intestinal immune homeostasis, or the reduction of inflammation [5,19].

In studies where *C. butyricum* was included in the diet of weanling piglets, diverse positive effects such as increased villus height and crypt depth, improved body weight, average daily gain and feed conversion rate were observed [38,63,64,65]. It also reduced the diarrhea rate [38,63,64,65]. Thus, the significantly higher abundance of this bacterium in the TG would suggest an overall better gut health condition in these slaughter pigs, although extrapolation of results from weanling piglets to adult pigs should be done with caution.

DNA-based studies have allowed detection of other uncultured microorganisms such as the *Bacteroidales* group S24-7 commonly found in homoeothermic animals and that appeared increased in the CG in this study (Table 2 and Figure 3). Although some murine studies have found fluctuations in its abundance in mice with different physiological conditions, its role on health is far from been determined [66]. More studies will be required to understand the role of these gut microorganisms.

## 5. Conclusions

The addition of PSB during the whole growing and fattening period at a dose of 3 kg per ton of feed did not significantly modify the overall richness of microbiota composition of pigs at slaughter, but significant changes in some specific taxa were detected. According to previous studies, most of the changes observed would be likely associated with better gut health parameters. An increase in the abundance of SCFA-producing strict anaerobes was detected (*Clostridium* and *Prevotella*) that may also help to reduce the presence of enteric pathogens. In any case, it should be noted that the role that a given taxon may have on gut health is likely influenced by the interactions of numerous variables, such as age, diet, environment, treatments, the presence of other taxa, infections or even the physiological status of the animal. Therefore, direct comparisons between studies are difficult.

## Figures and Tables

**Figure 1 animals-11-02137-f001:**
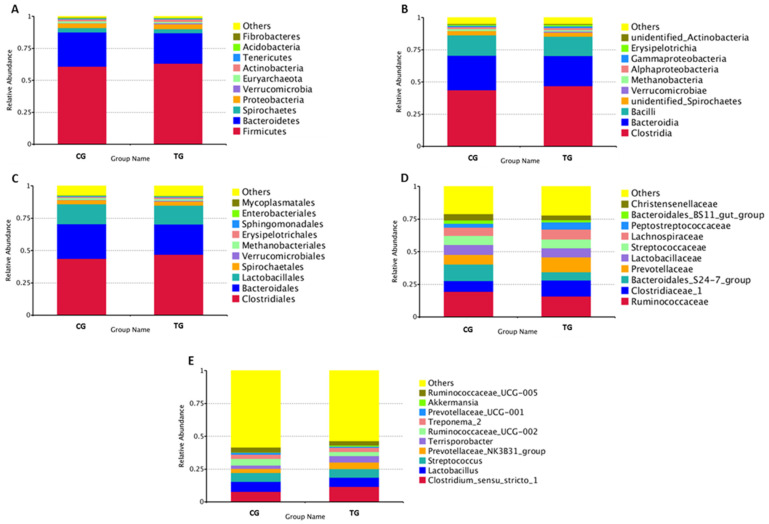
Percent relative abundance of the ten most abundant phyla (**A**), classes (**B**), orders (**C**), families (**D**), and genera (**E**) in the control (CG) and treatment (TG) group.

**Figure 2 animals-11-02137-f002:**
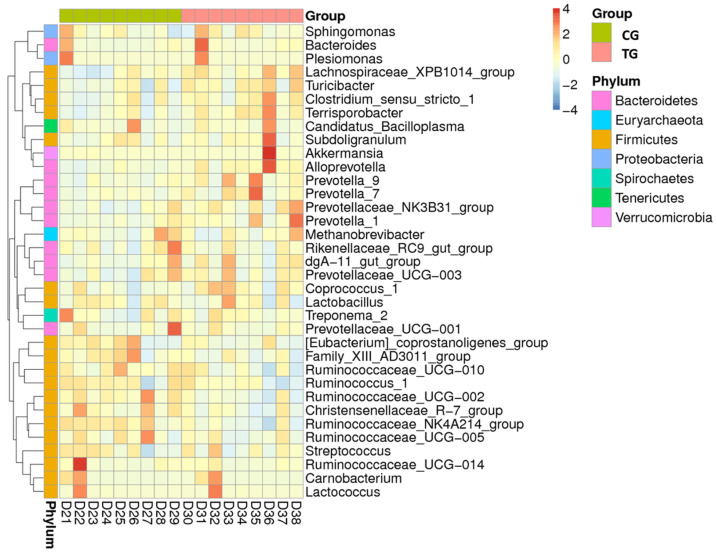
Heat map of relative differences of the 35 most common genera in the fecal samples of pigs fed a normal diet (CG) and a diet supplemented with protected sodium butyrate (TG). A value of −4 represents the disappearance of a particular genus, while a +4 value indicates that it increased from the initial value of 0.

**Figure 3 animals-11-02137-f003:**
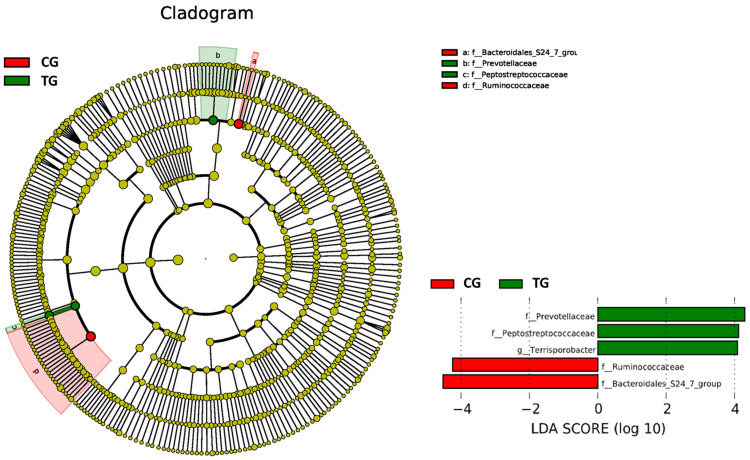
On the left side of the graph, differentially abundant microbial clades according to linear discriminant analysis effect size (LEfSe). On the right side, significant clades and their associated linear discriminant analysis (LDA) score for the control group (CG) and the treatment (TG) group are shown.

**Table 1 animals-11-02137-t001:** Effects of dietary protected sodium butyrate supplementation on the microbial alpha diversity in the colon content of growing-finishing pigs.

	Control Group		Treatment Group		
Index	Mean	SD *	Mean	SD	*p*-Value
Shannon	7.033	0.313	7.305	0.288	0.07
Simpson	0.968	0.014	0.975	0.0068	0.2
ACE	2549	369.23	2649	277.75	0.5
Chao1	2488	374.39	2582	289.37	0.5

* SD: Standard deviation.

**Table 2 animals-11-02137-t002:** *t*-test analysis of variation between groups. Differentially abundant taxa from fecal samples between the control (CG) and treatment (TG) group. Only the taxa that were significantly different between groups are listed.

Taxonomic Level	Taxon	CG		TG		*p*-Value
		Mean	SD	Mean	SD	
Phylum	*Deinococcus-Thermus*	4.5 × 10^−9^	2.8 × 10^−7^	8.8 × 10^−8^	4.6 × 10^−9^	0.032
Family	*Ruminococcaceae*	0.194	0.023	0.159	0.024	0.006
	*Bacteroidales* S24-7 group	0.127	0.066	0.064	0.037	0.027
	*Prevotellaceae*	0.075	0.051	0.114	0.039	0.047
	*Lachnospiraceae*	0.063	0.012	0.077	0.013	0.035
	*Peptostreptococcaceae*	0.030	0.014	0.055	0.023	0.015
	*Peptococcaceae*	0.001	0.3 × 10^−3^	0.002	0.4 × 10^−3^	0.026
Genus	*Terrisporobacter*	0.026	0.012	0.049	0.020	0.012
	*Ruminococcaceae* UCG-002	0.050	0.015	0.032	0.011	0.012
	*Alloprevotella*	0.006	0.003	0.014	0.009	0.030
	*Ruminococcaceae* NK4A214 group	0.021	0.003	0.015	0.003	0.003
	*Lachnospiraceae* XPB1014 group	0.012	0.004	0.018	0.004	0.016
	*Turicibacter*	0.005	0.002	0.008	0.002	0.033
	*Parabacteroides*	0.001	0.001	0.003	0.002	0.036
	*Eubacterium nodatum* group	0.001	0.3 × 10^−3^	0.002	0.001	0.027
	*Blautia*	0.002	0.001	0.003	0.001	0.042
	*Oscillospira*	0.004	0.001	0.003	0.001	0.016
Species	*Clostridium butyricum*	0.003	0.001	0.005	0.003	0.048

## Data Availability

Data available upon request.

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
