# Peer review of "Effects of Dietary Supplementation with Protected Sodium Butyrate on Gut Microbiota in Growing-Finishing Pigs"

_animals, 2021, doi:10.3390/ani11072137_

Round 1
Reviewer 1 Report
Include Latin names in italics throughout the article.
Materials and Methods
Add: housing conditions, dimension of pens,
Add: composition and nutritional characteristics of the diets, frequency of feeding, intake of the feed mixture, physical structure of the feed mixture, access to water, method of power supply,
106: consider listing a commercial name, add the type of oil.
Discussion
219: Table 2- add description statistics for each group (mean, SD, min., max.).
Conclusions
368: Rewrite, not cite other works.
Author Response
Add: housing conditions, dimension of pens
More information on this aspect has been added in the text. Since the study was carried out in a commercial pig farm, we have added a sentence indicating that "Animals were raised following European animal welfare regulations for pig farms (COUNCIL DIRECTIVE 2001/88/EC of 23 October 2001)".
Add: composition and nutritional characteristics of the diets, frequency of feeding, intake of the feed mixture, physical structure of the feed mixture, access to water, method of power supply
Since three different diets were administered during the whole period of fattening, we think it is much better to include this feed information in a supplementary table (S1)
106: consider listing a commercial name, add the type of oil.
We have added information regarding the type of vegetable fat (hydrogenated palm stearin)
219: Table 2- add description statistics for each group (mean, SD, min., max.).
We have included these data in the Table now.
368: Rewrite, not cite other works.
Done
Reviewer 2 Report
Review of the manuscript number: animals-1294560
In general, this interesting scientific article is very carefully written and documented with advanced statistical research methods. However, during the reviewing several questions have arisen.
- Line 100 - What was the n value for experimental and control groups -50:50?
- Line 105 – The details (composition) of diet components are missing. As described in many researches, the protein content, fibre, and resistant starch in the diet as well as the electrolyte balance could influence the fermentation products, thus modify the colonic microbiota composition. It was not written what nutritional program was used for this experiment and before experiment. No information about access to feed, water and housing conditions throughout the experimental period was found.
- Line 106 – Sodium butyrate used to the experiment (GUSTOR BP70, Norel S.A., Madrid, Spain), according to the manufacturer“…fat and sodium butyrate are part of the same matrix. This also ensures a gradual release along the digestive tract while coated products are liberated in the last parts of the gut…”It is known that absorption of free form (not coated) of sodium butyrate occurs mostly in the upper portion of the gastrointestinal tract improving intestinal morphology, intestinal permability and tight junction protein expression etc., limiting its actions in the large intestine. And when you used this product it is not clear how much sodium butyrate will reach the large intestine and can influence the colonic microbiota.
- Line 111 – What were the criteria for selecting pigs to evaluate the composition of the colon microflora? Was it a random selection? Was it an average gain or an average weight of 4 pens in each group? Was the diarrhea index monitored in the groups of the experiment?
- Line 141 – It is not clear what was the format of data presentation?
- In figure 1 the authors present some numerical values in control and experimental groups. These values should be supported by SD/SEM values.
- The Discussion is far too long. The authors should remove superfluous parts not directly related to the conducted experiment and focus on discussing the results presented in this manuscript. In the present form, the discussion did not throw any new light on the findings and significance of the results.
After making the appropriate improvements, this manuscript can be considered for further evaluation. As it stands, it cannot.
Author Response
In general, this interesting scientific article is very carefully written and documented with advanced statistical research methods. However, during the reviewing several questions have arisen.
Line 100 - What was the n value for experimental and control groups -50:50?
Now it is indicated that:
At slaughter, intestinal content from the colon of 18 randomly selected pigs (9 from the TG and 9 from the CG) was collected for laboratory analysis.
Line 105 – The details (composition) of diet components are missing. As described in many researches, the protein content, fibre, and resistant starch in the diet as well as the electrolyte balance could influence the fermentation products, thus modify the colonic microbiota composition. It was not written what nutritional program was used for this experiment and before experiment. No information about access to feed, water and housing conditions throughout the experimental period was found.
We have included a supplementary table with data on the three diests. Additional information on housing has been included as well
Line 106 – Sodium butyrate used to the experiment (GUSTOR BP70, Norel S.A., Madrid, Spain), according to the manufacturer“…fat and sodium butyrate are part of the same matrix. This also ensures a gradual release along the digestive tract while coated products are liberated in the last parts of the gut…”It is known that absorption of free form (not coated) of sodium butyrate occurs mostly in the upper portion of the gastrointestinal tract improving intestinal morphology, intestinal permability and tight junction protein expression etc., limiting its actions in the large intestine. And when you used this product it is not clear how much sodium butyrate will reach the large intestine and can influence the colonic microbiota.
We have modified the description of the product to make clear it was able to reach the distal part of the intestine:
This feed additive is a form of sodium butyrate (70%) part of which was protected with vegetable fat (hydrogenated palm stearin, 30%) in order to be able to reach the distal part of the intestinal tract, according to manufacturer.
Line 111 – What were the criteria for selecting pigs to evaluate the composition of the colon microflora? Was it a random selection? Was it an average gain or an average weight of 4 pens in each group? Was the diarrhea index monitored in the groups of the experiment?
We did not collect zootechnical parameters, so the pigs were randomly selected from each group. We have rewritten the sentence.
Line 141 – It is not clear what was the format of data presentation?
We have included now the following sentenceÇ:
Output data were stored in FASTQ format.
In figure 1 the authors present some numerical values in control and experimental groups. These values should be supported by SD/SEM values.
We present descriptive information previous to formal comparisons, as to get an overall picture of results. We think this is the standard way of presenting this type of results as observed in many other articles on this matter.
However, we have added this type of information in Table 2, where we show the comparisons that resulted significant.
The Discussion is far too long. The authors should remove superfluous parts not directly related to the conducted experiment and focus on discussing the results presented in this manuscript. In the present form, the discussion did not throw any new light on the findings and significance of the results.
We have reduced significantly the discussion and hopefully improved it
Reviewer 3 Report
Dear Authors,
I don't have any major suggestions, the minor ones are listed bellow.
Line 13 and elsewhere - Consider replacing “per Ton of feed” with “per ton of feed”.
Lines 41-42 - The sentence is difficult to read. Rephrase it as follows. “Gut microbiota is involved in many physiological functions, one of the most important being the digestion of nutrients.”
Lines 60 and 67 - Be consistent in your use of italics in bacterial taxonomy. For example, Proteobacteria are italicized in line 60 but not in line 67. The same is true for E. coli (lines 67, 74). Also, all phyla (genera, etc.) should be written the same way (italicized or not).
Line 95 - “limited period of time”
Line 127 - Consider replacing “According to the concentration . . . “ with “Depending on the concentration . . .”
Line 131-132 - Change “bright main strip” to “bright main band”. PCR strips are sets of small tubes.
Line 200-201 - Some spelling errors (“de”). Replace with “A value of -4 represents the disappearance of a particular genus, while a value of +4 indicates that it has increased from the initial value of 0.”
Line 227 - Replace “linear discriminate analysis” with “linear discriminant analysis”
Line 229 - Misspelled word. “tratment”
Lines 386, 387 - Hard to read sentence. Replace with “Ethical review and approval were waived for this study because no procedures were performed on live animals.”
Author Response
Line 13 and elsewhere - Consider replacing “per Ton of feed” with “per ton of feed”.
Done
Lines 41-42 - The sentence is difficult to read. Rephrase it as follows. “Gut microbiota is involved in many physiological functions, one of the most important being the digestion of nutrients.”
Done
Lines 60 and 67 - Be consistent in your use of italics in bacterial taxonomy. For example, Proteobacteria are italicized in line 60 but not in line 67. The same is true for E. coli (lines 67, 74). Also, all phyla (genera, etc.) should be written the same way (italicized or not).
Done
Line 95 - “limited period of time”
Done
Line 127 - Consider replacing “According to the concentration . . . “ with “Depending on the concentration . . .”
Done
Line 131-132 - Change “bright main strip” to “bright main band”. PCR strips are sets of small tubes.
Done
Line 200-201 - Some spelling errors (“de”). Replace with “A value of -4 represents the disappearance of a particular genus, while a value of +4 indicates that it has increased from the initial value of 0.”
Done
Line 227 - Replace “linear discriminate analysis” with “linear discriminant analysis”
Done
Line 229 - Misspelled word. “tratment”
Done
Lines 386, 387 - Hard to read sentence. Replace with “Ethical review and approval were waived for this study because no procedures were performed on live animals.”
Done